# Arabidopsis in the Wild—The Effect of Seasons on Seed Performance

**DOI:** 10.3390/plants9050576

**Published:** 2020-05-01

**Authors:** Deborah de Souza Vidigal, Hanzi He, Henk W. M. Hilhorst, Leo A. J. Willems, Leónie Bentsink

**Affiliations:** 1Wageningen Seed Science Centre Laboratory of Plant Physiology, Wageningen University, 6708 PB Wageningen, The Netherlands; dsvidigal@bejo.nl (D.d.S.V.); hzhe@mail.hzau.edu.cn (H.H.); henk.hilhorst@wur.nl (H.W.M.H.); leo.willems@wur.nl (L.A.J.W.); 2Bejo Zaden B.V., Trambaan 1, 1749 CZ Warmenhuizen, The Netherlands; 3College of Plant Science & Technology, Huazhong Agricultural University, Wuhan 430070, Hubei, China

**Keywords:** seed dormancy, seed longevity, field conditions, environmental effects, Arabidopsis

## Abstract

Climate changes play a central role in the adaptive life histories of organisms all over the world. In higher plants, these changes may impact seed performance, both during seed development and after dispersal. To examine the plasticity of seed performance as a response to environmental fluctuations, eight genotypes known to be affected in seed dormancy and longevity were grown in the field in all seasons of two years. Soil and air temperature, day length, precipitation, and sun hours per day were monitored. We show that seed performance depends on the season. Seeds produced by plants grown in the summer, when the days began to shorten and the temperature started to decrease, were smaller with deeper dormancy and lower seed longevity compared to the other seasons when seeds were matured at higher temperature over longer days. The performance of seeds developed in the different seasons was compared to seeds produced in controlled conditions. This revealed that plants grown in a controlled environment produced larger seeds with lower dormancy than those grown in the field. All together the results show that the effect of the environment largely overrules the genetic effects, and especially, differences in seed dormancy caused by the different seasons were larger than the differences between the genotypes.

## 1. Introduction

Seeds are part of the survival strategy of higher plants. Seeds allow the continuation of life after the mother plant senesces or is predated. Many types of plants, particularly the ephemerals, use seed dormancy as a survival strategy. Dormancy prevents seeds from germinating all at once after maturation and dispersal, thus avoiding the possible destruction of the species, in case of unpredictable and unfavourable environments after germination [1]. Seed dormancy contributes to the adaptation of plants to their environment by optimizing the germination to the right period of the year. The level of dormancy (primary dormancy) in seeds is usually determined by several factors, such as genetic and the environmental factors operating during development and maturation [1,2,3,4]. The same genetic and environmental factors that affect seed dormancy may also influence seed longevity, however in a reverse manner. This implies that increased seed dormancy co-occurs with reduced seed longevity and vice versa [5,6]. Seed longevity is defined as the viability of seeds after storage and mostly refers to storage in dry conditions [7]. The capacity to survive in the soil seed bank is another characteristic that is important for plant survival. Seeds shed by the mother plant, that do not immediately germinate, may enter the soil seed bank. In the soil, these seeds are able to adjust their dormancy state (secondary dormancy) in response to signals from the environment. The level of the response to the environment is largely affected by the same genetic factors that also determine their primary dormancy levels [8]. In the field, seeds are commonly subjected to fluctuating temperatures which usually comprise low night temperatures and high temperatures during the day. These diurnal temperature fluctuations are frequently effective in dormancy breakage of many species [1]. Seed dormancy can also be released by after-ripening (AR), which occurs by storing seeds in dry conditions during several months at a mild/warm temperature or by stratification, or a low-temperature treatment (cold stratification) of imbibed seeds [1,9].

*Arabidopsis thaliana* (Arabidopsis) is an annual species which is spread across the northern hemisphere and grows in contrasting habitats along latitude, longitude and altitude. This geographic variation may influence the life cycle of natural populations, which can be divided into winter and spring annuals [10]. The winter annual life cycle occurs when seeds germinate in autumn and the seedlings or rosettes are maintained during the winter, after which, during the spring or beginning of summer, the plants flower, set, and disperse seeds. Alternatively, the spring annual life cycle starts when seeds germinate in spring and grow into mature plants that flower, set seed and disperse their seeds in the same spring or summer [10]. The life cycle history of plants is important to the survival of the species because they need to avoid harsh environments and reproduce when the conditions are good enough for establishment and growth. This is especially important during germination and flowering [11]. For Arabidopsis it has been shown that the same genotypes might adopt winter or summer annual life strategies depending on the environment [11,12,13].

Environmental signals during seed development, maturation, and after dispersal, such as photoperiod, temperature, and drought stress, may influence the germination characteristics of the seeds at dispersal, affecting the requirements for dormancy to be relieved [10,14]. The effect can either be mediated by the mother plant and thus be purely maternal or be perceived by the embryo and therefore be zygotic [15]. The role of day length, light quality, temperature, water, and nutrients in determining the degree of dormancy has been investigated in a wide range of species [16,17,18]. The behavior of the progeny can be affected by day length during the last stages of seed maturation [2]. Germinability of some species can be higher for seeds produced under short days [19,20,21]. However, Contreras et al. [22] and He et al. [6] did not find any significant effect of day length on seed performance. Temperature during seed development and maturation is one of the most important determinants of seed germinability or seed dormancy in many species [6,16,23,24,25,26,27]. Generally seeds developed at warmer temperatures are less dormant at maturity than those that develop at cooler temperatures, as described for many species, including *Beta vulgaris*, *Lactuca sativa*, *Amaranthus retroflexus*, wild oat, *Avena fatua* [16], and Arabidopsis [6,26,28,29]. However, elevated maternal temperatures can enhance primary dormancy in some species such as *Syringa vulgaris* and *Syringa reflex* [30], *Sisymbrium officinale* [31]*, Xanthium pensylvanicum* and *Helianthus annuus* [18]. In Arabidopsis both seed yield and dormancy were highly reduced by higher temperatures, but accessions showed a differential response, demonstrating that this temperature response is not solely an environmental effect [27]. The natural genetic variation of the accessions also plays a role in seed performance phenology. This indicates that projected climate change may impact seed performance, but that the consequences will differ between species or within the same species. Insight regarding both the effect of the environmental conditions and the genetic basis is also important to understanding how traits like dormancy and longevity may have evolved.

It has been shown that both field and laboratory experiments may supply valuable knowledge about the regulation of germination and emergence in the field [10]. Here, we investigated the plastic response of seed performance (seed dormancy, seed longevity) and properties (seed size) on seeds of plants that have been grown in the field during all seasons of the year (spring, summer, autumn and winter) using a set of near isogenic lines (NILs) that contain introgression fragments which are known to affect seed dormancy and seed longevity.

## 2. Results

### 2.1. Environmental Conditions during the Seasons

To investigate the influence of seasons on seed performance, we performed a field experiment using eight different Arabidopsis genotypes. Two wild types, Landsberg *erecta* (L*er*) and Columbia (Col-0) and five Near Isogenic Lines (NIL*DOG1*, NIL*DOG2*, NIL*DOG3*, NIL*DOG6*, NIL*DOG22*) [32], as well as a mutant with a lesion in the *DOG1* gene (*dog1-1*) were sown in the field in the consecutive seasons of 2012 and 2014. These genotypes are known to have different dormancy and longevity levels when seeds are produced in controlled conditions [5,6,32]. After seed harvest, seed size, seed dormancy, and seed longevity levels were investigated. The conditions that the plants experienced during the different seasons were determined by monitoring, air temperature, the soil temperature (data not shown), precipitation, day length and average sun hours (Figure 1). Both soil and air temperatures gradually increased during the spring, autumn, and winter experiments, while they gradually decreased for plants of the summer experiment. During the spring experiment, the day length increased from 14:17 h to 16:44 h per day, whereas in summer, the day length decreased from 14:32 to 9:17 h. The autumn and winter experiments included the shortest day of the year, with only 7:43 h of light. During these short days plants were in the rosette (autumn) and seedling (winter) stage. The reproductive period of these plants started when the day length increased again. Seeds were harvested at a day length of 16:38 h for autumn and winter experiments. On average 2014 was warmer then 2012, furthermore the summer of 2012 was wetter than that of 2014 (Figure 1).The growing season largely affected the duration of the life cycle, therefore the time between seed sowing and harvesting was recorded for every season. Plants that were sown during autumn and winter had the longest life cycles (234 and 183 days in 2012 and 196 and 112 days in 2014, respectively; Figure 1). In both cases this was caused by the fact that these plants overwintered in the vegetative state and only started flowering in spring. This also indicates why seeds of both of these growths were harvested in June of the next year (2013 and 2015, respectively). Spring and summer sown plants had shorter life cycles, with 71 and 87 days in 2012 and 69 and 87 days in 2014 (Figure 1).

### 2.2. Influence of Seasons on Seed Performance

We assessed seed performance of plants that were grown in the field and compared the effect of the different seasons and the different years. Plants from seeds that were sown during winter showed a higher number of branches, greater plant height and higher seed yield (data not shown). The other three seasons (spring, summer and autumn) displayed similar plant phenotypes. Seed performance over the seasons varied. Generally the smaller seeds that were harvested in the summer season displayed the highest dormancy levels (higher number of days of seed dry storage required to reach 50% germination (DSDS50) values, thus longer after ripening (AR) time) and the shortest seed longevity (Figure 2). Moreover, *dog1*-*1*, NIL*DOG22* and NIL*DOG3* produced the smallest seeds, especially in the 2014 experiment (Figure 3A, Appendix A). Spring, autumn and winter experiments produced seeds with very similar dormancy and longevity levels (Figure 2B). Unfortunately, the dormancy data (DSDS50 values) of the summer of 2014 are missing, therefore no conclusions can be drawn on possible dormancy differences between the two summers. The individual genotypes responded differently to the environments (Figure 3B, Appendix A). In control conditions there were large differences in primary dormancy and longevity among these genotypes [5,6,32]. In the field, these differences are less obvious, indicating that the environment strongly affected these traits. We observed that the low dormancy genotypes NIL*DOG2*, NIL*DOG22* and *dog1-1* also are the lower dormant genotypes in the field. Moreover, in controlled conditions, dormancy and seed longevity are negatively correlated [5]. This effect was also observed in the field, especially when we look at the averages of the seasons. The dormancy levels of NIL*DOG2* seeds were lower in spring 2012 compared to the winter and autumn experiments. This genotype by environment interaction was however not detected in 2014. The *dog1-1* mutant is a non-dormant genotype and the seasons did not affect the very low DSDS50 values (Figure 3B, Appendix A). However, we could see an effect of the seasons, especially in summer, when we investigated the after-ripening behaviour. We found that *dog1-1* seeds harvested from the summer experiment required 76 and 63 days in 2012 and 2014, respectively, to completely ripen (Appendix A).

## 3. Discussion

### 3.1. Seed Performance in Response to Different Seasons

Seeds are responsible for the development of the next generation, and how these seeds perform is determined by both the genotype and the environment of the mother plant [6,33]. For many species, adjusting the life cycle is a strategy to avoid sub-optimal environmental conditions and maximize survival and fitness of later stages of growth [34]. Seedling emergence is usually synchronized with seasonal changes in the environment [3,18]. Earlier developmental stages, such as seedlings, are expected to be more sensitive to sub-optimal conditions than later stages, i.e., vegetative plants [29]. However, it is a huge challenge to predict the effects of a changing environment on natural populations. Understanding the response of plants to their environment or more specifically to different seasons is also relevant for agriculture. Climate change might impact crop management and, e.g., result in earlier crop sowing [35]. Our study provides insight into the effects that growing seasons have on seed germination performance. To reduce the complexity, we focus our discussion on findings described for Arabidopsis. Moreover, we compare our findings in detail to those of He et al., [6] since this study made use of the exact same genotypes grown in controlled environmental conditions.

Life cycle strategies largely vary over the seasons, with that of plants sown during autumn and winter being the longest. For the autumn the life cycle took 234 days in 2012 and 196 days 2014 and for the winter 183 days in 2012 and 112 days for 2014 (Figure 1). Both the autumn and winter plants survived the winter in the vegetative state and started flowering in the following spring, which resulted in more or less the same harvest times for both seasons. The delay of plant growth in these seasons is probably due to the low temperature, at which the risk of mortality or flower abortion is high [36]. Plants that are sown in spring and summer have a comparable life cycle length (71 and 87 days in 2012 and 69 and 87 days in 2014 respectively, Figure 1). These values are comparable to the life cycles of the same genotypes when grown under controlled conditions, which took 72 days from sowing to seeds harvest [6]. Life cycle strategies are normally determined by controlling the timing of germination and flowering [12,37]. In our experiment, germination was controlled by us since we performed artificial stratification that allows a synchronized uniform germination of all genotypes. Nevertheless, we see different life cycle strategies as seeds sown in spring and summer represent the typical spring life cycle in which seeds germinate, whereby plants grow to maturity, flower, set seed, and disperse their seeds in the same spring or summer season. Seeds sown in autumn and winter follow a winter life cycle in which seeds germinate in autumn, spend winter in the vegetative state, to flower and disperse seeds in next spring (Figure 1). Moreover, the genetic material used did not vary much for flowering time except for the NIL*DOG2* lines, due to the presence of the CRYPTOCHROME 2 (CRY2) Cvi allele. Under controlled conditions these lines flower slightly earlier than wild type L*er* [38,39].

The seeds produced by plants that were sown in summer and harvested in autumn displayed the most divergent seed performance (smaller seeds, higher dormancy and reduced longevity) compared to the other seasons (Figure 2). Seed longevity of the summer grown plants significantly differed between the summers of 2012 and 2014. This can be the result of the higher temperatures at seed harvest in 2014 (20 °C) than that of 2012 (17 °C), as well as an effect of the shorter storage time (2500 and 1700 days for 2012 and 2014, respectively). Seeds of the summer growth were harvested when day length, sun hours, and temperature started to decrease (Figure 1). It is known that the photoperiod during seed maturation is a predictable indicator of the season [21]. Earlier it has been found that low light intensity and short days, as well as high temperature during seed maturation, resulted in smaller seeds [6]. The same study showed that photoperiod did not have any significant effect on seed dormancy, but low light intensity and low temperature enhanced the dormancy levels. In our study both temperature and day length of the summer harvests was different from that in the other seasons, suggesting a role both factors on seed dormancy. The day length of the summer harvests was 10 h for the summer growth compared to 16 h in spring, autumn and winter respectively. The proposed effect of the photoperiod is in agreement with a report by Postma and Agren [40]. These authors demonstrated that differences in dormancy levels of seeds from two different field sites, one in Sweden and another in Italy, are likely determined by differences in the maternal photoperiod since the average temperatures between the fields did not differ much. Also, Munir et al. [21] suggested that the maternal photoperiod may contribute to variation in dormancy levels in Arabidopsis depending on progeny stratification. These authors showed that for seeds matured under short days, germination percentage and rate increased in stratified seeds but were lower in unstratified ones. The same group demonstrated that seeds matured under short days displayed slightly later germination than seeds matured under long days, but the germination percentage was higher in seeds matured under short day conditions than seeds matured under long days [37].

Temperature at seed harvest over the seasons varied between 10 and 18 °C. The temperature at harvest of the summer growth was the lowest, however the average temperature during seed maturation did not differ from that during the other seasons. It is known that low temperature promotes deep primary dormancy, whereas warm temperatures reduce dormancy [6,11,26,27,41]. For Arabidopsis, it was shown that a difference of 1 °C can have a consequence for seed dormancy [13]. The induction of high dormancy at the end of summer might be important to avoid germination in autumn, and the cold experienced during the winter would ensure germination in the next spring when temperatures become favourable for germination [1,42]. According to Donohue [10], germination is influenced by pre-dispersal seed maturation conditions and post-dispersal seasonal conditions. Therefore, both the influence of the environment during seed development and the link with the environmental conditions during seed germination are important for the establishment and depth of dormancy.

The importance of both temperature and photoperiod has also been demonstrated by Montesinos-Navarro et al. [34] who simulated spring and autumn germination conditions in a population of Arabidopsis genotypes. For autumn germination, seeds were kept moist with progressive reduction of temperature and hours of light (which is equivalent to our summer experiment). In the spring germination conditions seeds were kept dry and in the dark to simulate autumn, followed by 30 days at 4 °C to simulate winter, then moistened and exposed to a progressive increase of temperatures and hours of day light (which is comparable to our winter experiment). These authors found that germination behaviour was significantly different in autumn versus spring simulated conditions, in which seeds exposed to autumn simulated conditions, displayed a decrease of 33% in final germination and a decrease in germination speed as compared to spring germination conditions.

### 3.2. Genotypic Response to the Seasons

Most genotypes showed similar responses to the different seasons. For all of these, the summer experiment resulted in the highest dormancy level. However, in 2012, NIL*DOG2* showed a strong interaction with the environment: in the spring experiment it displayed the lowest dormancy level, followed by winter, autumn and summer, respectively (Figure 3). Phenotypic plasticity among different genotypes is likely caused by environment interactions [6]. This is supported by genetic variation for maternal photoperiod and seasonal dormancy that was found when analysing a recombinant inbred line population [21]. Moreover, it is thought that post-dispersal seasonal conditions influence germination responses even more than photoperiod during seed maturation [37]. However, it has been shown that secondary dormancy levels induced after dispersal correlate with primary dormancy levels induced during seed maturation [8].

Strategies for breaking dormancy are genotype dependent as well. In Col-0 dormancy can be broken by a short cold treatment, suggesting that the maturation of this genotype occurs in cool days of autumn and the seeds germinate in the following spring. However, L*er* seeds need a warm period followed by a cold treatment in order to germinate, which suggests that the seeds mature in autumn, will not germinate in spring but only in the following autumn, because the seeds need to experience the warm temperatures of summer [43]. Variation in germination among natural accessions of Arabidopsis was found for seeds matured at low temperatures, but not in seeds that matured at high temperature [41]. In a study of 18 wild accessions of Arabidopsis, differences in response to chilling (simulation of winter) for seeds maturated in different temperatures (20, 15, and 10 °C) was shown. Some accessions did not respond to a cold treatment in the dark, independent of the temperature of maturation, although low temperatures during seed maturation were more effective to increase the dormancy levels in most of the accessions [28].

### 3.3. Field Versus Controlled Conditions

We have grown the same set of genotypes in four seasons in two different years and identified largely similar effects on seed performance in these two years even though it is known that variation in temperature, rainfall, photoperiod and nutrients under field conditions normally hamper repeatability of experimental results. To understand the effect of the environment on seed performance we have compared our results to that of seeds of the same genotypes that were produced in controlled conditions (20 °C/18 °C (day/night) under 16 h photoperiod of artificial light (150 μmol·m^−2^·s^−1^) [6]. We noticed that plants in controlled conditions produced larger seeds which were less dormant than seeds that were produced in the field, independent of the season or genotype (Figure 2 and Figure 3). We did not compare seed storage data for the field and control conditions since, for the control experiment, longevity assessments were performed using artificial ageing. It is known that a short exposure of plants to extreme environments such as heat, stress or drought during seed filling can decrease seed set, seed size, seed weight, reduce yield and also result in low seed quality [44]. In the field such changes of the environment can occur suddenly, especially fluctuations in temperature. To interpret the differences in seed size, seed dormancy and longevity between seeds developed in the field and under controlled conditions we present three hypotheses. The first one is the lack of fluctuations in temperature and photoperiod during the complete plant life cycle in the controlled conditions, resulting in large seeds with low dormancy. This is the opposite of what normally occurs in nature where fluctuations of the environment are prevalent during the life cycle of plants, which may cause deep dormancy as a survival mechanism against (short-term) changes in the environment. It is not known how alternating (diurnal) temperatures during plant growth influence seed dormancy. Only few studies have attempted to understand the physiological process by which alternating temperatures appear to be important for germination in dormant species, such as increased germination in the dormant ecotype Cvi of Arabidopsis [45] and various circadian clock mutants [46]. Indirect effects through the mother plant have been demonstrated for Arabidopsis, alternating the growth temperature of the mother plant before and after flowering affected seed dormancy [26]. Postma and Agren [40], studying three different maternal environments (one greenhouse environment and two field sites), found differences in dormancy levels among the environments. The greenhouse environments led to lower seed dormancy than the two field environments. This may be related to the temperature inside the greenhouse that was higher compared with the field sites.

The second hypothesis assumes that the combination of different environments causes differences. It is known that single environments, as described for temperature and photoperiod, may influence seed performance, especially seed dormancy, but a combination of environments may show a different response. It was reported that a combination of phosphate and temperature did not have a significant effect on plant and seed performance, while the individual environments, phosphate, or temperature, did show an effect [6].

The third hypothesis to explain the difference in seed performance of plants grown under controlled conditions and in the field assumes the existence of nutrient effects. Under controlled conditions, nutrients were provided to the mother plant every two days as a standard nutrient solution [6], whereas in the field experiment, no additional nutrients were supplied to the plants. The nutrient composition and levels of the soil were not analysed. The addition of nutrient fertilizer to parental plants is known to decrease dormancy in the seeds of several species [16]. However, Arnold et al. [47] observed that reduced potassium nutrition in developing seeds of *Sorghum bicolor* increased the germinability because the ABA content of the seeds was reduced. Conditions favoring nitrate accumulation in mother plants of Arabidopsis resulted in lower dormancy [6,48]. Matakiadis et al. [49] reported that high nitrate concentrations may release seed dormancy in Arabidopsis, in part by reducing abscisic acid levels. A similar effect was found by Modi and Cairns [50] who observed that molybdenum deficiency in wheat resulted in lower seed dormancy by decreasing abscisic acid content. It is likely that a deficiency of molybdenum, a co-factor of nitrate reductase, reduces nitrate reductase activity, resulting in higher nitrate levels of the seeds. Phosphate levels did not influence dormancy, but increasing phosphate and phytate content increased germination under stress conditions [6].

## 4. Material and Methods

### 4.1. Plant Materials and Growth Conditions

The field experiment was executed in a field at Wageningen University (51°59′05.7″ N, 5°39′29.2″ E, 25 m above sea level), The Netherlands. Arabidopsis genotypes (Landsberg *erecta* (L*er*), Columbia (Col-0) and other genotypes with the L*er* genetic background (5 near isogenic lines (NILs)), namely NIL*DOG1*-Cvi (Cape Verde Islands), NIL*DOG2-*Cvi, NIL*DOG3*-Cvi, NIL*DOG6*-Kas-2 (Kashmir), and NIL*DOG22*-An-1 (Antwerpen) [32,38], and the *dog1-1* mutant [51] were subjected to the field conditions spanning all seasons of the year (spring, summer, autumn, and winter). Plants were grown in 2012 and 2014 spring (between April and July), summer (between August and November), autumn (between October and June) and winter (between December and June). They are respectively referred to as spring, summer, autumn, and winter even though part of the life cycle occurred in another season (Figure 1).

Seeds were sown in trays, one row per genotype followed by a 4-day cold treatment at 4 °C, and transferred to a plastic tunnel without any environmental control to avoid possible damage by heavy rain. The trays were watered every two days for approximately two weeks until seedlings were fully established for spring, summer and autumn experiments. The winter experiment in 2012 took 14 weeks because of the low temperatures that slowed down seed germination and subsequent seedling growth. Germinated seedlings were transferred to different trays (28 seedlings per genotype per tray with four replicates (trays) per genotype). These trays were kept in the tunnel for one more week, to ensure the survival of seedlings. After that, the trays were transferred to the field. Sensors to measure the soil temperature were evenly distributed over the trays and temperature data were recorded every 10 min. Data of air temperature and average sun hours per month was obtained from the weather station at De Bilt, The Netherlands, (http://www.knmi.nl/ klimatologie /daggegevens/selectie.cgi) and data regarding day length were obtained from (http://www.sunrise-and-sunset.com/nl/nederland/amsterdam).

Seeds were harvested when the plants had stopped flowering and more than 50% of the siliques were brownish. The seeds were harvested as a bulk from each tray.

### 4.2. Seed Performance Analyses

Seed size was analysed by taking photographs of the seeds on white filter paper (20.2 × 14.3 cm white filter paper, Allpaper BV, Zevenaar, The Netherlands, http://www.allpaper.nl) using a Nikon D80 camera fixed to a repro stand with a 60 mm macro objective. The camera was connected to a computer with Nikon Camera Control Pro software version 2.0. Clustering of seeds was prevented as much as possible. The photographs were analysed using ImageJ (http://rsbweb.nih.gov/ij/) by combining colour thresholds (Y100-255U0-85V0-255) with particle analysis that automatically scored seed size as the area of selection in square millimetres.

Germination experiments were performed as described previously [52]. In brief, two layers of blue germination paper were equilibrated with 47 mL demineralized water in plastic trays (15 × 21 cm). Six samples of approximately 50 to 150 seeds were spread on wetted papers using a mask to ensure accurate spacing. Piled up trays were wrapped in a closed transparent plastic bag. The experiment was carried out in a 22 °C incubator under continuous light (143 μmol·m^−2^·s^−1^). Pictures were taken twice a day for a period of 6 days using the same camera and software as described for seed size. Germination was scored using the Germinator package [52]. To measure the seed dormancy level (DSDS50; days of seed dry storage required to reach 50% germination), germination tests were performed weekly until all seed batches germinated for more than 90% [6]. For the summer experiment of 2014, after-ripening was only followed for 91 days after harvest. Because of this, DSDS50 values could not be calculated. To show that both summer experiments show largely the same after-ripening behaviour, the germination percentages of the individual experiments are presented (Appendix A). Seed longevity was measured by germinating the seeds 2500 and 1700 days after harvest for the 2012 and 2014 experiments, respectively. Maximum germination percentage after five days of seed imbibition was recorded.

## 5. Conclusions

Our study indicates that, depending on the time of the year when plants are grown, seed performance can be different, but this is relatively stable over the two years that we studied. Plants grown during summer, when the days began to shorten and the temperature started to decrease, produced smaller seeds with deeper dormancy and reduced longevity. Seed dormancy was largely controlled by the environment seen the fact that the differences caused by the different seasons are larger than the differences among the genotypes. In addition, plants grown in a controlled environment produced larger seeds with lower dormancy than those grown in the field. Overall, we conclude that environmental factors largely overrule the genetic differences.

## Figures and Tables

**Figure 1 plants-09-00576-f001:**
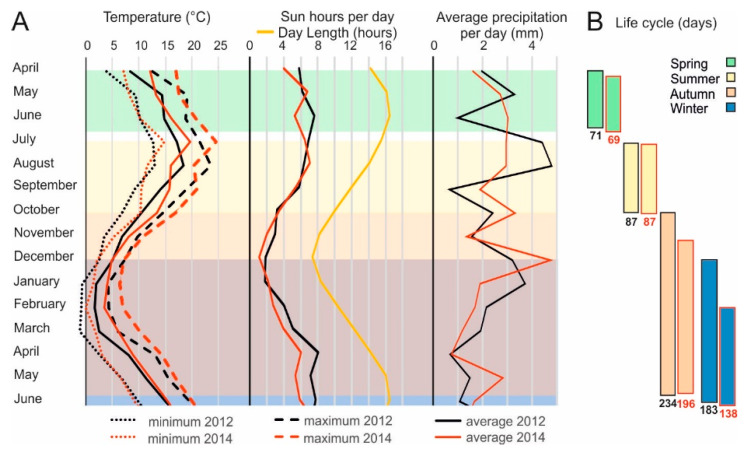
Environmental conditions and their effect on the plant life cycle. (**A**) The minimum, average and maximum air temperature, sun hours per day (average per month), day length (yellow line) and average precipitation per day during four seasons in 2012 (black lines) and 2014 (red lines) at the location where the plants were grown. (**B**) The plant life cycle from seed sowing to seed harvest of each experimental season (spring, summer, autumn and winter, in 2012 (black outlines) and 2014 (red outlines)).

**Figure 2 plants-09-00576-f002:**
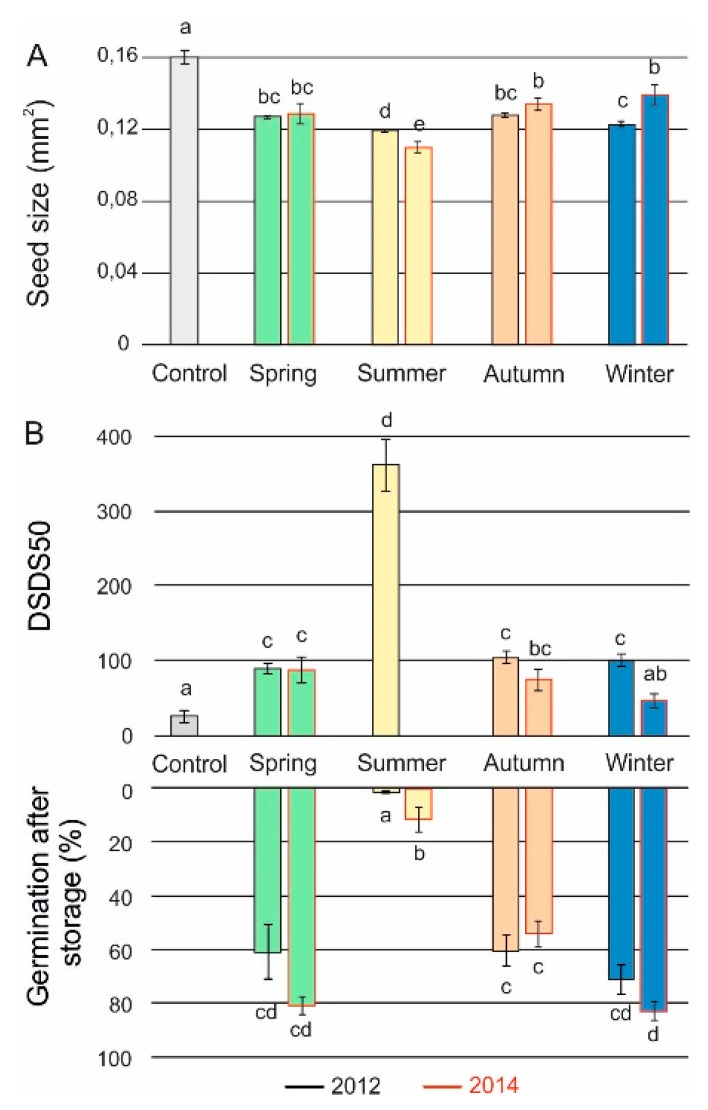
Average seed performance over the different seasons. (**A**) Average seed size (mm^2^) and (**B**) Average of dormancy (DSDS50) and seed germination after storage (longevity) levels of plants grown in different seasons (spring, summer, autumn and winter) in 2012 (black outline) and 2014 (red outline). The grey bars represent the data of the control environment as published by He et al. [6]. Averages of four replicates are displayed. Error bars represent standard error. Means followed by the same letter did not differ by Tukey’s test (P < 0.05).

**Figure 3 plants-09-00576-f003:**
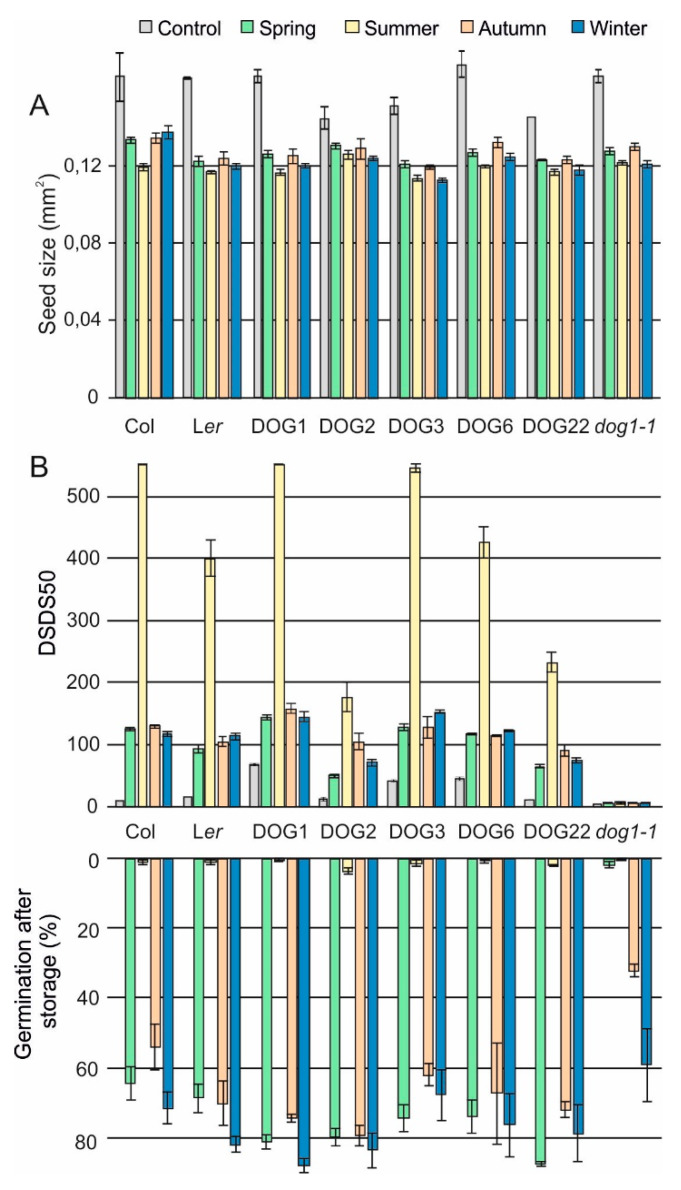
Seed performance of the individual genotypes over four season. (**A**) Seed size (mm^2^), (**B**) Seed Dormancy (DSDS50) and seed germination after storage (longevity) levels for each genotype for all seasons of 2012. The grey bars represent the data of the control environment as published by He et al. [6]. Averages of four replicates are displayed. Error bars represent standard error.

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
