# Peer review of "Arabidopsis in the Wild—The Effect of Seasons on Seed Performance"

_plants, 2020, doi:10.3390/plants9050576_

Round 1

Reviewer 1 Report

In the manuscript ‘Arabidopsis in the Wild; the Effect of Seasons on Seed Performance’, the authors present their work on differences for seed dormancy caused by the different seasons were larger than the differences between the genotypes.

The authors performed a field experiment using eight different Arabidopsis genotypes. The genotypes were grown in different seasons. The authors measured seed size, seed dormancy and seed longevity of the harvested seeds.

This work is highly suggestive and provides an important contribution to understanding of seasonal effects on plants.

I have a comment:

The authors showed the seasonal effects on seed performance. Readers in this journal might be interested in biological importance of the seasonal effects. Detailed explanation is necessary on this point.

Author Response

Reviewer 1:

Comments and Suggestions for Authors

In the manuscript ‘Arabidopsis in the Wild; the Effect of Seasons on Seed Performance’, the authors present their work on differences for seed dormancy caused by the different seasons were larger than the differences between the genotypes.

The authors performed a field experiment using eight different Arabidopsis genotypes. The genotypes were grown in different seasons. The authors measured seed size, seed dormancy and seed longevity of the harvested seeds.

This work is highly suggestive and provides an important contribution to understanding of seasonal effects on plants.

I have a comment:

The authors showed the seasonal effects on seed performance. Readers in this journal might be interested in biological importance of the seasonal effects. Detailed explanation is necessary on this point.

Authors response: In the original text we address this in the abstract line 13-14. “To examine the plasticity of seed performance as a response to environmental fluctuations, eight genotypes known to be affected in seed dormancy and longevity were grown in the field in all seasons of two years.”

In the discussion we added an explanation more specifically related to the biological aspects of the seasons. Line 176/179. “Understanding the response of plants to their environment or more specifically to different seasons is also relevant for agriculture. Climate change might impact crop management and e.g. result in earlier crop sowing (Estralla et al., 2009). Our study provides insight into the effects that growing seasons have on seed germination performance.”

Reviewer 2 Report

The authors present a complex and elaborate study design. They investigated the seed performance of eight Arabidopsis genotypes in the open field over several years. This is really an effort!

The fact that in the second experiment release of dormancy could be followed for only until 91 DAH is not a severe lapse (and is due to the time-consuming study design, I guess) since data of the first experiment could be reproduced regarding this period of time.

However, the second winter experiment is not clear to me. In the first winter experiment, plants were grown from december to june. In the second experiment, plants were grown from march to june in the same year (Fig. 1). So is the latter experiment not rather a "spring experiment" than a "winter experiment"?

Author Response

Reviewer 2:

Comments and Suggestions for Authors

The authors present a complex and elaborate study design. They investigated the seed performance of eight Arabidopsis genotypes in the open field over several years. This is really an effort!

The fact that in the second experiment release of dormancy could be followed for only until 91 DAH is not a severe lapse (and is due to the time-consuming study design, I guess) since data of the first experiment could be reproduced regarding this period of time.

However, the second winter experiment is not clear to me. In the first winter experiment, plants were grown from december to june. In the second experiment, plants were grown from march to june in the same year (Fig. 1). So is the latter experiment not rather a "spring experiment" than a "winter experiment"?

Authors response: This is a very good observation of the reviewer. We went back to the data and noticed that the date in the graph was the planting date instead of the sowing date. For most of the experiment the seeds germinated shortly after sowing, however in winter 2014 it took almost one month. We have corrected the bar in Figure 1B. This correction did not completely solve the issue mentioned by the reviewer. Both the autumn and winter experiment in 2014 started a bit later then in 2012. This has a logistics related explanation. The plot in the field did had a limited space and we could only start new experiments after cleaning out the earlier ones, this was unfortunately a bit delayed in 2014. We do however not worry about this since it did not affect the timing of seed production.

Reviewer 3 Report

The authors investigated how seed performance changes when Arabidopsis seeds were grown in different seasons in fields and concluded that the environment overrules the genetic variation especially for seed dormancy. I appreciate the presented work. I need some clarifications and also have few suggestions to improve the manuscript.

  1. It would be very informative to have correlation coefficient plots between parameters to see which factors have correlations. I felt the authors try to emphasize that there is a negative correlation between the seed size and the degree of dormancy. In addition, in the discussion, there are many factors described and it is difficult to follow what messages the authors to try to convey. Therefore, a figure (can be a supplementary figure) showing correlation coefficient plots and cite this in the manuscript will help readers understand the content in a much higher level.
  2. The authors utilize the previous work (He et al.), but the actual data are not presented in this manuscript. It is important to show the current work data and the previous data work together like in a table format, for example, so that readers understand further.
  3. In the discussion section, there are many work cited and described. However, I do not clearly see the summary or actual discussion of how these published work help us understand the data the authors obtained. Discussion section should be rearranged.

Minor points:

  1. In Fig.1, how was the average sun hours per day obtained? It was not clear.
  2. In Fig.1. A and B are missing in the figure, but mentioned in the legend.
  3. In Fig.1, why are the life cycles of 2014 autumn and winter significantly shorter than those of 2012?
  4. Line 138-140 is a discussion and should be incorporated into the discussion section.
  5. Line 147: Why is it remarkable? In Line 242-244, the authors explain a bit, but it is not clear to me why this is so remarkable. More discussion is required.
  6. Discussion 3.1: It is not clear to me about the relationship between temperatures and photoperiods. The author stated that even 1oC difference can affect seed dormancy. But at the same time, Line 203-207 indirectly mentions that the temperature effect is minimum. More clarification is required.
  7. In comparison between Fig. 3 and Fig. S1, why germination rate of dog1-1 spring differs between 2012 and 2014?
  8. In comparison between Fig. 3 and Fig. S1, why germination rate of DOG2 summer differs between 2012 and 2014?
  9. It would be good to explain the conceptual difference between DSDS50 and After-ripening requirement so that readers can understand the authors' point easily.
